# Assessing the Post-Activation Performance Enhancement of Upper Limbs in Basketball Athletes: A Sensor-Based Study of Rapid Stretch Compound and Blood Flow Restriction Training

**DOI:** 10.3390/s24144439

**Published:** 2024-07-09

**Authors:** Shuang Cui, Zhihao Du, Nannan Wang, Xiuli Zhang, Zongquan Li, Yanping Zhang, Liang Wang

**Affiliations:** 1School of Physical Education, China University of Mining and Technology, Xuzhou 221116, China; kdcuishuang@126.com (S.C.); dzh_cumt_edu@163.com (Z.D.); lizongquan658200@126.com (Z.L.); 2School of Physical Education, Ningbo University, Ningbo 315000, China; 3School of Physical Education, Zhengzhou University, Zhengzhou 451000, China; 4Wuhan First People Hospital, Wuhan 430000, China; 17629207306@126.com

**Keywords:** rapid stretching compound training, blood flow restriction training, post-activation performance enhancement, basketball players, upper limb muscle

## Abstract

Objective: This study introduces a novel methodology combining rapid stretch compound training with blood flow restriction (BFR) to investigate post activation performance enhancement (PAPE) in basketball players, a field that has been predominantly explored for lower limbs. We aimed to assess the efficacy of this combined approach on upper limb muscle performance in athletes. Methods: We employed a randomized, self-controlled crossover trial with ten male basketball players. The bench press throw (BPT) served as the primary metric, with players undergoing four interventions post-baseline: (1) STR—plyometric training; (2) BFR—blood flow restriction; (3) COMB—STR integrated with BFR; and (4) CON—control. Innovatively, we utilized an intelligent tracking sensor to precisely measure peak power (PP), peak velocity (PV), mean power (MP), and mean velocity (MV) at 4, 8, and 12 min post-intervention, providing a detailed temporal analysis of PAPE. Results: The COMB intervention demonstrated superior PAPE effects at 4 min, significantly outperforming STR and BFR alone and the control group across all measured indices (*p* < 0.05). Notably, the COMB group maintained these improvements for PV, PP, and H up to 12 min post-intervention, suggesting a prolonged effect. Conclusion: (1) The COMB stimulation has been shown to successfully induce PAPE more effectively than STR and BFR modality alone. (2) It appears that the optimal effects of PAPE are achieved within 4 min of exercising under this COMB. By the 12 min mark, only the COMB group continued to show significant improvements in PV, PP, and H compared to both the baseline and the CON group, while the effects in the STR and BFR groups further diminished. This suggests that although the PAPE effect is maintained over time, its optimal performance may peak at the 4 min mark and then gradually weaken as time progresses.

## 1. Introduction

Post-activation performance enhancement (PAPE) refers to a physiological phenomenon where a brief, pre-existing submaximal or maximal intensity exercise intervention enhances muscle performance during subsequent training sessions [1]. Currently, most experimental studies on PAPE focus on lower limb movements [2], and it has been generally confirmed that short-term high-intensity warm-up exercises can enhance athletic performance [3], such as jumping performance [4], and significantly improve lower limb explosive strength [5]. Common training methods to induce PAP, such as high load (80% 1RM) deep squats, have significantly improved athlete’s vertical jump and sprinting performance [6]. However, a disadvantage is that it is more challenging to balance the relationship between exercise fatigue and the PAPE gain effect caused by large load interventions, and PAPE occurrence cannot be guaranteed [7].

Blood flow restriction (BFR) training refers to the use of a binding cuff at the proximal end of the limb to completely block the flow of venous blood while reducing but maintaining arterial blood flow to the muscles, thereby causing the formation of a venous pool [8]. BFR training utilizes a low-intensity load intervention to facilitate the occurrence of PAPE while better avoiding the issue of exercise fatigue. Earlier studies, such as Moore et al. (2004), found that when instructing subjects to perform upper-extremity single arm elbow flexion and extension with an occlusion pressure of 100 mm HG and exercise intensity set at 50% 1RM, a 51% PAPE was observed. Wilk’s research team varied the exercise modality, subjecting participants to a bench press training regimen that entailed three sets of three repetitions at an intensity of 70% of their one-repetition maximum (1RM), with occlusive pressure set at 90% of the individual’s arterial occlusion pressure (AOP). They discovered that, irrespective of blood flow restriction (BFR) intervention, the second group exhibited significantly higher peak power (PP) and peak velocity (PV) during bench press compared to the first group, indicating the occurrence of PAPE. However, it is noteworthy that the magnitude of PAPE effect was significantly greater in the compression group than that in the non-compression control group (Wilk et al., 2020) [9]. This probably indicates that compression training can generate higher neuromuscular stimulation, resulting in better recruitment of high-threshold motor units, which is more conducive to the successful induction of the PAPE effect (Larsen et al., 2021) [10].

In basketball, the PAPE effect was mainly studied in the lower limbs, and it has been found that conditioning activity with 80% of 1RM as the high-load exercise intervention can significantly improve lower limb performance. However, there are fewer studies on the upper limb. Given the unique characteristics of basketball, the movement mode in basketball involves coordination between the upper and lower limbs, and upper limb muscle strength is crucial for dribbling, passing, and physical confrontation in basketball. Therefore, inducing upper limb muscle recruitment and the PAPE effect is as important as that for the lower limbs. Current research on inducing the PAPE effect in the upper limb has primarily been conducted through the bench press throw (BPT) test, but the results have been inconsistent. Liossis et al. (2013) [11] observed that after five bench press sessions at different intensities, a significant increase in BPT performance (PAPE effect) was observed at 4 min after low-intensity (65%1RM) bench press sessions and continued until the eighth minute, but only at the eighth minute after high-intensity (85% of 1RM) bench press sessions. This suggests that medium- and high-intensity bench press sessions can trigger PAPE, and the timing of the PAPE effect induction is related to the exercise intensity, and the level of fatigue caused. However, Chinese scholar Shi Haipeng has stated that preliminary maximal isometric voluntary contractions lasting either 3 or 5 s fail to evoke upper limb PAPE effects in basketball players. Given the coexistence of PAPE effects and fatigue, this may be related to exercise-induced fatigue during the process in which the Contraction-Associated Potentiation (CA) induces PAPE. Whether PAPE induction is successful depends on the balance between enhancement and fatigue.

In summary, PAPE has predominantly been explored in the context of lower limb muscle performance in sports science, with a relative dearth of research on its impact on upper limb musculature. This is particularly true for sports like basketball, which require the integrated strength of both upper and lower body segments. BFR training has shown promise as an effective method for inducing PAPE without increasing the risk of fatigue, thereby offering a potential strategy for enhancing athletic capabilities. The purpose of this study is to address the significant gap in understanding the effects of combined rapid stretching and BFR training on the upper limbs of basketball athletes. In light of this, the present study aims to investigate the impact of combining rapid stretching compound training with BFR training on the upper limb PAPE effects in basketball athletes. The goal of this study is to provide new scientific evidence and methods for the training of basketball athletes. The contributions of this study are as follows: (1) It is the first to systematically evaluate the impact of rapid stretching compound training combined with blood flow restriction training on upper limb PAPE in basketball players. (2) Through a randomized controlled trial, it was found that this training protocol significantly improved the athletes’ performance at the 4 min mark, offering a new perspective for enhancing muscle strength and explosive power. (3) The findings not only enrich the scientific knowledge of sports training, but also provide valuable references for the training practices of basketball players, holding significant scientific and practical prospects.

## 2. Research Objects and Methods

### 2.1. Objects of Study

This study calculated the number of participants through G*Power software 3.1 version, and finally selected ten participants. From 10 to 15 October 2022, this experiment enrolled ten basketball players from the School of Physical Education at China University of Mining and Technology as participants. All subjects were adults who volunteered for the study and had signed the informed consent form. Their basic information is presented in Table 1. Prior to the experiment, the subjects were informed about the purpose of the study, the experimental implementation method, and the potential risks involved during the experiment. The consent of both the coaches and the basketball players themselves was obtained. The training movements involved in this experiment, and the interventions that distinguished it from regular training, were explained to the subjects before the test. Each subject was given the opportunity to wear the BFR equipment for acclimation training one week prior to the experiment. This experiment was conducted in strict accordance with the Helsinki Declaration, and was reviewed and approved by the Ethics Committee Member of Basic Medical School of Zhengzhou University (ZZUIRB2022-JCYXY0017). The experiments took place at the Basketball Fitness Center of China University of Mining and Technology, Institute of Physical Education.

### 2.2. Research Methodology

#### 2.2.1. Experimental Design and Intervention Program

Subjects were recruited one week before the start of the experiment, and all participants underwent measurements for height, weight, and acromial grip distance. Individual verbal reports of bench press 1RM were also recorded. Subsequently, the bench press 1RM test was conducted three days prior to the official test, based on the individual verbal reports of 1RM data.

Experimental Equipment: The present study utilized the Theratools BFR (Blood Flow Restriction) device, which consisted of a pressure pump and a compression cuff. The cuff was fastened with Velcro and had a width of 7.5 cm. The compression was applied to the upper arm, specifically the middle to the proximal third of the upper arm near the heart side. The adopted occlusion pressure was 30 mmHg, with a compression pressure of 140 mmHg (moderate occlusion pressure). During testing, an intermittent flow restriction stimulation method was used, where pressure was applied during the exercise phase and was released during the rest interval between sets a group [12].

On the day of the formal experimental test, subjects warmed up using the same standard warm-up procedure as the bench press 1RM test, followed by performing an upper body explosive strength test (pre-test). The testing procedure was consistent with that used in previous studies [13]. Subjects were scheduled to perform three sets of two repetitions each of the BPT at 30% of 1RM. Twenty minutes after completing the BPT, the exercise intervention was performed. The experimental group consisted of three different interventions: the STR group, which performed three sets of five repetitions each of rapid extension compound training with high-five push-ups, bouncing push-ups, and supine medicine ball push-ups; the BFR group, which performed blood flow restriction training with an occluding pressure of 140 mmHg only; and the COMB group, which trained using fast extension compound training with high-five push-ups, bouncing push-ups, and supine medicine ball forward push-ups, along with blood flow restriction stimulation at 140 mmHg. The CON group engaged in sedentary rest. The intervals between sets of blood flow restriction push-up training were 2 min for each mode, and the time interval between different training programs was 72 h. This was conducted in order to prevent muscle damage caused by resistance training from affecting the subjects’ test state and to reduce the interference effect between different modes of pre- and post-testing [14]. The specific process is illustrated in Figure 1. At 4 min, 8 min, and 12 min after the aforementioned exercise intervention, subjects were instructed to perform the BPT post-test experiment using the exact same pre-test procedure. This allowed for the assessment of upper extremity explosive strength after the exercise intervention [6]. All subjects served as their own controls. The total duration of a single experiment was approximately 90 min.

#### 2.2.2. Experimental Tests

(1) Bench Press 1RM Maximum Strength Test

Each subject performed the supine flat bench press 1RM test three days prior to the official experiment. To ensure standardization of the bench press 1RM test, all subjects followed the same requirements, as follows: they were in the supine position during the preparation process, with the head, shoulders, and hips maintained in close contact with the bench; they held the barbell with a closed, rotated forward grip and used the most comfortable grip spacing for their individual needs. The barbell should touch the chest on the way down. The 1RM bench press test began 5 min after the warm-up, with the participant first bench-pressing 80% of their 1RM using the verbally reported load and then resting for 2 min before starting the incremental bench press. Testers had the load increased by 5–10 kg and the subjects were instructed to bench press the 1RM at the first attempt to lift.

During the bench press 1RM test, subjects were first asked to warm up according to the standard warm-up procedure, as follows: first, a 5 min treadmill jogging warm-up session, followed by a 5 min bench press-specific warm-up, with each subject performing 15, 10, and 5 repetitions of the bench press at 20%, 40%, and 60% of the verbally reported bench press 1RM. Finally, a 3 min retractor stretch was performed on the pectoralis major, triceps, and deltoids of the involved upper extremity muscle groups. The 1RM bench press test began 5 min after the warm-up, with the participant first bench-pressing 80% of the verbally reported 1RM for 3–4 repetitions and then resting for 2 min before starting the incremental bench press. Testers first used 80% of the 1RM load based on the increase in load 5–10 kg, immediately after the participant was instructed to carry out the second bench press test, complete 2–3 repetitions after a break of 2 min, and continue to repeat the incremental load of 1–2 times bench press training. The testers had the load increased by 5–10 kg, and the subjects were instructed to bench press the 1RM at the first attempt to lift, and, if successful, then continue to repeat the incremental bench press. If this was successful, the subject continued to repeat the incremental bench press, and if this was unsuccessful, the load was reduced by 2.5–5 kg and the subject tried the bench press 1RM again. The 1RM of the bench press for all subjects was determined in 5–6 experiments [5]. The 1RM of the bench press was determined in 5–6 trials for all subjects.

(2) Rapid Scaling Compound Training Test

The following movements are set according to National Strength & Conditioning Association [15].

① High-five push-ups

The action steps are as follows (Figure 2). (1) Standard push-up posture: feet together, arms apart, distance slightly wider than shoulder width, and abdominal tightening to maintain a straight back without collapsing the waist. (2) Controlled maneuver to make the body quickly fall, and then reversing the maximum force to quickly push upward, while keeping the feet on the ground. (3) During the process of quickly pushing upward, the body hangs in the air while performing a rapid high-five action, and then returns to the standard push-up position. Repeat the same sequence with a 1 s break between each high-five push-up and repeat across a total of three sets of five repetitions.

② Bouncing push-ups

The action steps are as follows (Figure 3). (1) Assume a standard push-up posture: feet together, arms apart, with a distance slightly wider than shoulder width, and tighten the abdomen to maintain a straight back without collapsing the waist. (2) Control the body to quickly descend, then reverse the maximum force to rapidly push upward, ensuring that the feet remain in contact with the ground. (3) As the body quickly pushes upward, it suspends in the air, the arms are rapidly extended and fall to the sides of the pedals, followed by a return to the standard push-up ready stance. Rest for 1 s between each pop-up push-up, and then proceed to repeat the aforementioned movement steps. Complete a total of three sets of five repetitions.

③ Supine medicine ball forward push

Action steps are as follows (Figure 4). (1) Assume a supine position, knees bent, feet naturally open, and hold a 5 kg medicine ball on the chest, with the tester standing above the subject ready to catch the ball. (2) Maintain a ready supine posture, with the back tightly pressed against the yoga mat, and then rapidly push the medicine ball forward to its highest point using maximum force in a controlled manner. (3) The tester catches the falling medicine ball and returns it to the subject. A 1 s interval is observed between each supine forward push of the medicine ball, after which the aforementioned steps are repeated. A total of three sets of five repetitions of this maneuver are performed.

(3) Bench Press Test

The bench press test was conducted using the same methodology as Ferreira S. L. et al. (2012) [16]. The test protocol was identical to that of Team Ferreira S. L., with a load arrangement of 30% of the subject’s bench press 1RM, with the test equipment being a Smith machine. At the beginning of the test, subjects were instructed to assume a flat bench press supine position, maintaining the same barbell grip and grip distance as in the bench press 1RM test. Additionally, a tester was positioned to protect one side of the bench press. Once all preparations were complete, the tester lifted the barbell to a position where the elbows were slightly straight, and then rhythmically controlled the barbell to slowly descend to a point 3–5 cm above the chest line; subsequently, the barbell was rapidly pushed upwards with maximum force, with the hands releasing at the highest point (West et al., 2013) [17]. After the barbell was thrown, the tester was responsible for timely control of the barbell to ensure the safety of the subject (GARCÍA-RAMOS et al., 2018) [18]. The accuracy and practical application of the Vmaxpro sensor for assessing exercise speed and load speed variables have been recognized by scholars, effectively evaluating kinematic indicators such as movement speed (Dragutinovic et al., 2024) [19]. Moreover, its ability to effectively judge the validity of kinematic indicators (r = 0.92–0.99, SEE = 0.02–0.13 m/s) ranks second among four linear sensing devices for monitoring athletic performance (Gymware (1st), Push (3rd), Flex (4th)). Therefore, in this study, recording test indicators through Vmaxpro has high reliability and validity (Fritschi et al., 2021) [20]. The changes in peak power (PP), peak velocity (PV), mean power (MP), and mean velocity (MV) during the BPT were recorded using a German intelligent tracking and sensing device (Vmaxpro). Refer to Figure 5 for further details.

### 2.3. Statistical Methods

Before analysis, we conducted normality tests on the data. Since two-way repeated-measures analysis of variance (ANOVA) typically assumes a normal distribution of data, we used the Shapiro–Wilk test to examine the distribution of data for each indicator under different conditions. The results showed that the data distribution for most indicators under different conditions did not conform to a normal distribution (*p* < 0.05). Therefore, we performed a logarithmic transformation on the raw data to meet the assumption of normal distribution. The transformed data passed the normality test, ensuring the validity of the ANOVA results.

Statistical analysis of the upper limb explosive strength index data in the BPT test before and after the exercise intervention was performed using SPSS 26.0. The data are presented as mean ± standard deviation (M ± SD). A repeated-measures two-way ANOVA (exercise mode × rest time) was used to statistically analyze the explosive strength indices PV, MV, PP, MP, and H in the BPT test before and after the intervention with different exercise modes. When the interaction was significant, pairwise sample *t*-tests were conducted within groups, and one-way repeated-measures ANOVAs were performed between groups. The significance level for difference testing was set at *p* < 0.05.

## 3. Findings

### 3.1. Multiple Comparison Analysis

In this study, a two-way repeated-measures analysis of variance (ANOVA) was employed to assess the effects of different exercise modes on upper limb explosive strength indices (PV, MV, PP, MP, and H), as well as the main effect of time and the interaction between group and time. The significance level was adjusted to *p* < 0.0033 after applying the Bonferroni correction to reduce the risk of type I errors due to multiple comparisons (Table 2).

The analysis results revealed that the four exercise groups (STR, BFR, COMB, and CON) had a highly significant impact on PV (F(1, N) = 6.366, *p* = 0.004, η**2 = 0.097), MV (F(1, N) = 9.573, *p* = 0.000, η**2 = 0.134), and MP (F(1, N) = 10.037, *p* = 0.000, η**2 = 0.137), and a significant impact on PP (F(1, N) = 3.277, *p* = 0.014, η**2 = 0.051). The main effect of time was not significant for any of the indices (all *p* > 0.05), indicating that the influence of rest time on upper limb explosive strength was limited. Furthermore, the interaction between group and time was only significant for the PV index (F(1, N) = 2.790, *p* = 0.041, η**2 = 0.043), suggesting that different training groups had varying impacts on PV over time.

Effect sizes (Cohen’s d) were calculated (according to Cohen’s criteria, 0.2 is a small effect, 0.5 is a medium effect, and 0.8 is a large effect) to further elucidate the moderate to large effects of the four exercise groups (STR, BFR, COMB, and CON) on PV, MV, and MP (d = 1.5, 1.9, 2.0), and a medium effect on PP (d = 0.8). These findings indicate that variations in exercise groups have a substantial impact on improving upper limb explosive strength, particularly for strength–speed indices.

### 3.2. Effects of Each Mode of Exercise Intervention on Changes in Upper Extremity Explosive Strength Indices at Different Time Points (Table 3)

PV (Peak Velocity): At 4 min post-test, the BFR group and COMB group values were significantly higher than the pre-test values. Additionally, the BFR group and COMB group values were also significantly higher than the STR group value. Both BFR and COMB values were significantly higher than the CON group values. At 8 min, only the STR group value was significantly higher than the pre-test value, with no significant differences observed in the other groups. At 12 min, the COMB group value was significantly higher than the pre-test value, and was also significantly higher than the blank CON group value.

MV (Mean Velocity): At 4 min, the MV values of the BFR group and COMB group were significantly greater than the pre-test value. The COMB group performed the best, with significantly greater values than the STR group, BFR group, and COMB group. At 8 min, the STR group and COMB group values were significantly greater than the pre-test values. The STR group value was significantly greater than the BFR group and COMB group values. At 12 min, only the COMB value was significantly greater than the pre-test value, with no significant differences between the other groups.

PP (Peak Power): At 4 min, both the BFR and COMB group values were significantly larger than the pre-test value. The BFR and COMB group values were also significantly larger than the COMB group value. Additionally, the COMB group value was significantly larger than the STR group value. At 8 min, only the STR group value was significantly larger than the pre-test value. There was no statistically significant difference between the groups at 12 min.

MP (Mean Power): At 4 min, the BFR group and COMB group values were significantly greater than the pre-test values. The BFR group value was also significantly greater than the CON group value. The COMB group had the best performance of MP, which was significantly greater than that of the STR group, BFR group, and COMB group. At 8 min, the STR group and COMB group values were significantly greater than the pre-test value. The STR group value was also significantly greater than the BFR group and the COMB group values. There was no statistically significant difference between the groups at the 12th minute.

**Table 3 sensors-24-04439-t003:** List of changes in the evaluation indices of the explosive strength of the upper limbs.

Group	Time	PV (m/s)	MV (m/s)	PP (W)	MP (W)	H (cm)
Pre-test	4 min	2.18 ± 0.142	1.25 ± 0.088	923.6 ± 146.48	433 ± 76.413	35.97 ± 8.149
	8 min	2.22 ± 0.138	1.26 ± 0.079	922.3 ± 133.34	436.75 ± 75.621	37.8 ± 8.356
	12 min	2.23 ± 0.128	1.26 ± 0.057	956.7 ± 131.32	444.5 ± 68.129	38.71 ± 6.003 &
STR group	4 min	2.24 ± 0.14 &#	1.27 ± 0.082 &	934.75 ± 141.37 &	445.1 ± 70.508 &	40.85 ± 6.723 *
	8 min	2.27 ± 0.141 *	1.33 ± 0.066 *	970.7 ± 136.83 *	465.2 ± 66.972 *	44.67 ± 9.706 *
	12 min	2.25 ± 0.141	1.29 ± 0.08	961.25 ± 134.25	456.11 ± 63.576	42.15 ± 6.367
BFR group	4 min	2.39 ± 0.272 *	1.31 ± 0.053 *&	976.05 ± 146.45 *	454.67 ± 68.579 *&	44.14 ± 7.644 *
	8 min	2.27 ± 0.138	1.28 ± 0.062 ∆&	946 ± 148.92	446.95 ± 62.402 ∆	41.12 ± 6.108 &
	12 min	2.23 ± 0.135 §	1.29 ± 0.067	940.39 ± 128.07	450.56 ± 59.615	40.18 ± 8.360 &
COMB group	4 min	2.32 ± 0.077 *	1.34 ± 0.031 *	981.05 ± 138.03 *	466.22 ± 55.941 *	45.8 ± 6.786 *
	8 min	2.27 ± 0.125	1.32 ± 0.039 *	949.35 ± 128.81	458.66 ± 60.513 *	44.82 ± 5.757 *
	12 min	2.30 ± 0.128 *	1.31 ± 0.058 *	969.63 ± 140.05	461.3 ± 61.932	45.53 ± 6.204
CON group	4 min	2.18 ± 0.119 &	1.28 ± 0.07 &	926.7 ± 131.43 #&	436.88 ± 68.678 &#	39.11 ± 7.937 *∆&#
	8 min	2.23 ± 0.132	1.29 ± 0.067 ∆	935.07 ± 138.19	445.6 ± 73.417 ∆	38.98 ± 7.444 ∆&
	12 min	2.23 ± 0.139 &	1.29 ± 0.054	935.13 ± 128.75	451.03 ± 60.34	40.3 ± 6.837 &

Note: * Indicates a significant difference in the change in upper extremity explosive strength indices between pre-intervention (pre-test) and post-intervention BPT tests; ∆ indicates a significant difference in the change in upper extremity explosive strength indices in the STR group compared to the other groups; # indicates a significant difference in the change in upper extremity explosive strength indices in the BFR group compared to the other groups; & indicates a significant difference in the change in upper extremity explosive strength indices in the COMB group compared to the other groups; and § indicates a significant difference in the change in upper extremity explosive strength indices within each group. Significant differences in the changes in upper limb explosive strength indices before and after different times. A significant difference was defined as *p* < 0.05; peak power (PP), peak velocity (PV), mean power (MP), and mean velocity (MV).

## 4. Discussion and Analysis

### 4.1. Analysis of the Effect of Upper Extremity PAPE Induction after Rapid Stretching Compound Training Intervention

The STR group’s plyometric training induced a significant PAPE effect at 8 min post-intervention, as shown by increased PV, MV, PP, and MP indices, highlighting the effectiveness of plyometric training in enhancing upper limb explosive strength. Furthermore, at 12 min post-intervention, there were no significant differences in the indices within the STR group, suggesting that the PAPE effect had subsided. Therefore, the study concluded that plyometric training can effectively induce an upper limb PAPE effect at the 8 min mark. Finlay et al. (2022) [21] classified different modes of regulated activity, including bench press variations, specific exercises (improved bar throw, bat swing specific, cable pulley, elastic resistance, and bodyweight exercises), and self-weight activities by incorporating 31 studies. They found a significant enhancement in acute exercise performance in several specific exercise combinations. The study results showed that, after 8–12 min of recovery following a bench press at least 80% of one’s one-rep maximum, the power output of the trajectory bench press can be significantly increased by 30–40%. Additionally, the superheavy barbell throw in specific exercises can increase the throwing distance by about 1.7–8.5% after 3 min of recovery. The light-weight bat swing specific to the sport and the isometric contraction specific to the swing result in a subsequent game-weight bat swing speed increase of approximately 1.3–4.9%. This indicates that plyometric training can promote the recruitment of motor units, thereby enhancing strength development. The findings of this study are consistent with previous research, confirming that upper limb plyometric training can significantly improve a subject’s explosive strength and power output.

However, different studies have offered varying opinions on the optimal timing for inducing a PAPE effect with plyometric training. Ulrich et al. (2017) found [22], in their study on upper limb plyometric training, that bench press strength significantly increased at 8 min post-training, indicating that the PAPE effect was successfully induced. In contrast, Wei Hongwen et al. (2022) observed [23], in their study on lower limb training, that the explosive strength of the lower limbs in the experimental group was significantly improved at 4 min post-training. The results of this study support the findings of Ulrich et al., indicating that the upper limb PAPE effect is induced after 8 min, which is inconsistent with the results of Wei Hongwen et al.’s lower limb training study. This discrepancy may be due to several factors: First, both this study and Ulrich’s study focused on upper limb training and used the supine throw as the test exercise, while Wei Hongwen’s study concentrated on lower limb training, which may be the main reason for the difference in the timing of the PAPE effect. Second, Asadi et al. (2016) pointed out that basketball players may gain greater training benefits from plyometric training than athletes of other types [2]. Considering that the subjects in this study were basketball players, while Wei Hongwen’s subjects were university students majoring in physical education, the difference in specialties may be another factor affecting the timing of the induction of the PAPE effect. In summary, this study and the related literature collectively support the conclusion that plyometric conditioning stimulus can effectively induce the PAPE effect, but the occurrence time of the upper limb PAPE effect (8 min) is later than that of the lower limb PAPE effect. This finding is of significant importance for developing targeted training programs and for optimizing athletes’ physical performance.

### 4.2. Analysis of the Effect of Upper Extremity PAPE Induction after Blood Flow 

#### Restriction Training

Our findings indicate significant increases in explosive strength metrics (PV, MV, PP, and MP) 4 min post-intervention compared with pre-test values, with PV and PP also being notably higher than the CON group. These data suggest that an interventional protocol involving compression without exercise can significantly enhance upper limb explosive strength in subsequent BPT tests at the 4 min mark, effectively inducing PAPE and outperforming the effects seen with rapid stretch-shortening cycle exercises. Two possible explanations for this phenomenon include the findings of da Silva Novaes et al. (2021) [24], asserting that pre-exercise IPC increases stores of ATP and creatine kinase, delays acidosis, and reduces exercise-induced fatigue, thereby improving muscle function. In addition, Valenzuela et al. observed through thermographic imaging that skin temperatures of the exercised muscle groups, biceps brachii, and pectoralis major were significantly elevated post-compression intervention [25]. It has been previously stated that the occurrence of PAPE correlates with post-exercise increases in muscle temperature [26], implying that compression-induced enhancement in hemodynamic response elevates muscle temperature higher under compression than in non-compressed environments, which could be more conducive to PAPE induction.

However, despite the BFR group demonstrating significantly higher PV values at the 4 min mark compared to the STR group, MV and MP were significantly lower at the 8 min mark; by the 12 min mark, differences between groups were not significant. Thus, the efficacy of solely using compression stimuli to induce PAPE is evident for less than 8 min. This opposes previous studies, such as that of Wei Hongwen et al. (2022) [23] which did not elicit lower-limb PAPE effects in a group undergoing only compression regardless of the timing [23]. Variability between studies may relate to differences in test areas. The current experiment used upper limb compression and analyzed PAPE using BPT, while the aforementioned study used lower limb compression and assessed PAPE with vertical jump tests. Furthermore, research has suggested that the induction of the PAPE effect may be related to the level of neuromuscular adaptation of the subject’s Type II muscle fibers [27]. Due to the higher potential for phosphorylation of the regulatory light chain (RLC) in Type II muscle fibers, this may enhance the activity of the peripheral neuromuscular system in the spinal cord, thereby improving the efficiency of muscle force production and facilitating the induction of PAPE. In this study, the subjects were basketball players, and considering the high demands of basketball on upper limb explosive power, such as rapid passing and shooting, these actions require a high degree of neuromuscular coordination and effective activation of Type II muscle fibers. Therefore, compared to university students with general training levels, basketball players may be more efficient in activating Type II muscle fibers, which could be a key factor in the successful induction of the PAPE effect in this experiment.

### 4.3. Analysis of the Effect of Upper Extremity PAPE Induction after Rapid Stretching Compound Training Combined with Blood Flow Restriction Stimulation

In our study, the STR experimental group, which received a combination of rapid contrasting stretches and compression stimuli, demonstrated significantly elevated PV and PP values 4 min post-intervention compared to pre-test and control levels. The MV and MP showed the greatest increase, significantly exceeding all groups’ values, including the pre-test, STR, BFR, and CON groups. The findings at the 4 min timepoint suggest that the rapid contrast stretching combined with compression stimulation is the most effective method of inducing PAPE effects among all the examined approaches. Current research, both domestic and international, indicates that compression training produces a better PAPE response than non-compression training methods. International researchers like Doma et al. (2020) found that [28], after participants with resistance training experience performed lunge squat exercises in three sets of eight repetitions, compression training led to significant increases in jump height (~4.5% ± 0.8%) and flight time (~3.4% ± 0.3%) from within 6 to 15 min post-exercise, indicating successful PAPE induction. A similar pattern was not noted in jump performance measurements following non-compressive interventions on the lunge exercise. The research team of Wilk found that the compression group showed superior improvements in peak velocity, peak power, average velocity, and mean power during subsequent testing of press exercises when using 90% of Arterial Occlusion Pressure (AOP) for three sets of three repetitions compared to the non-compressive group [28]. Studies reported that BFR stimuli may increase the recruitment in fast muscle fibers by inducing quicker fatigue in slow muscle fibers due to intramuscular hypoxia, and rapid recruitment of high-threshold motor units is one of the mechanisms contributing to PAPE [1]. This suggests that compression training in this experiment could have further facilitated the occurrence of PAPE through such mechanisms [29].

Additionally, the experimental results indicate a diminishing PAPE effect at the 8 min mark and a resurgence at the 12 min mark when compared to the 4 min performance. Studies, such as those by Loenneke et al. (2009) [30], found that BFR during training, using methods like banding, could lead to an accumulative fluid in limbs, insufficient venous return, and an optimal acidic environment for muscle activation due to increased intracellular pH, promoting post-unrelease enhanced hemodynamic response and energy substrate delivery to the muscles. At the 8 min mark, fatigue accumulation may overtake the PAPE effect until a possible resurgence between the 8–12 min window. Previous research highlights a “window period” for PAPE effects, during which regulation activity intensity may elicit higher force levels if adequate, but dominant fatigue might weaken performance, followed by rapid fatigue dissipation, allowing PAPE to regain dominance within a “second window period” (Pan et al., 2021) [31]. Experiments with compression training indicated that metabolic stress during such a routine might lead to muscle neural disorder, resource depletion, metabolite accumulation, decreased energy metabolism rate, and thus greater fatigue compared to non-compressive training, with larger-scaled compressive pressure correlating with more adverse effects on training efficacy [32]. In summary, this study concludes that rapid stretch-shortening exercise combined with compression stimuli intervention produces the best induction of the PAPE effect at the 4 min mark, with waning effects between the 8–12 min mark due to the limited-time existence of the PAPE effect. Thus, selecting this training mode for inducing upper limb PAPE effect should consider a longer recovery time post the initial testing at 4 min (>8 min) before performing the second test.

### 4.4. Limitations

There are still some limitations in this study. In this study, the assessment of 1RM was conducted three days prior to implementing the PAPE protocol and pre-existing delayed onset muscle soreness (DOMS) was not evaluated prior to the PAPE tests. However, it is important to note that the participants in this study were high-level basketball players who regularly engage in high-intensity training, indicating a higher level of muscle adaptability and potentially lower sensitivity to DOMS. Furthermore, a three-day interval between the 1RM assessment and the PAPE tests is considered to be sufficient to assume that any DOMS resulting from the assessment would have likely subsided. This hypothesis is indirectly supported by the fact that no athletes reported experiencing pain or discomfort during the PAPE tests.

Additionally, our experimental design focused on specific time points (4 min, 8 min, and 12 min) to assess the PAPE effect, without exploring the effects over a longer time frame, which may limit a comprehensive understanding of the duration of PAPE.

### 4.5. Significance and Value

The study systematically evaluated, for the first time, the effects of combined fast stretching complex training with BFR training on upper limb PAPE in basketball players. Through a randomized controlled trial, the research found that this protocol significantly improved athletic performance in basketball players at the 4 min time point, indicating the potential advantages in promoting muscle strength and explosiveness. This finding provides a novel training strategy for basketball players, especially those needing to enhance upper limb strength and speed before training or competition. Furthermore, the research results shed light on the physiological mechanisms underlying muscle adaptive enhancement in sports training, offering new experimental data for the theoretical development of sports training science and exercise physiology. In conclusion, this study not only enriches the scientific knowledge in sports training, but also provides valuable references for the training practices of basketball players, demonstrating high scientific significance and practical prospects.

### 4.6. Future Work

This study offers fresh perspectives on the impact of integrating rapid stretching compound training with blood flow restriction on post-activation performance enhancement in basketball players’ upper limbs. Despite these contributions, the scope for future research remains vast. Future directions should include expanding the diversity and size of the sample to encompass a wider range of genders, ages, and training backgrounds, thereby broadening the applicability of our findings. Longitudinal studies are needed to assess the sustained effects of PAPE training on athletes’ long-term performance and adaptive capabilities. Additionally, comparative analyses of various PAPE training protocols will be conducted, examining factors such as training intensity, duration, and recovery periods. The cross-application of PAPE training to other sport disciplines will be explored to determine its suitability for diverse athletic demands. Innovation in training methodologies will involve leveraging cutting-edge sports science technologies like wearable devices and biofeedback to refine PAPE training and evaluation techniques. Finally, there is a need to further integrate theoretical understanding with practical application, enhancing the development of training strategies that are both scientifically grounded and practically effective.

## 5. Conclusions

(1) The COMB stimulation has been shown to successfully induce PAPE more effectively than STR and BFR modality alone. (2) It appears that the optimal effects of PAPE are achieved within 4 min of exercising under this COMB. By the 12 min mark, only the COMB group continued to show significant improvements in PV, PP, and H compared to both the baseline and the CON group, while the effects in the STR and BFR groups further diminished. This suggests that although the PAPE effect is maintained over time, its optimal performance may peak at the 4 min mark and then gradually weaken as time progresses.

## Figures and Tables

**Figure 1 sensors-24-04439-f001:**
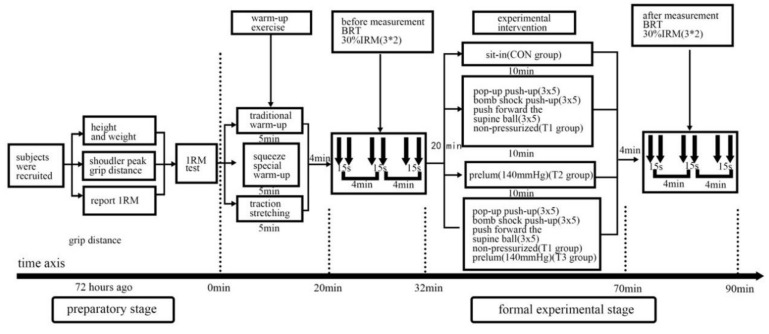
Experimental flow chart.

**Figure 2 sensors-24-04439-f002:**
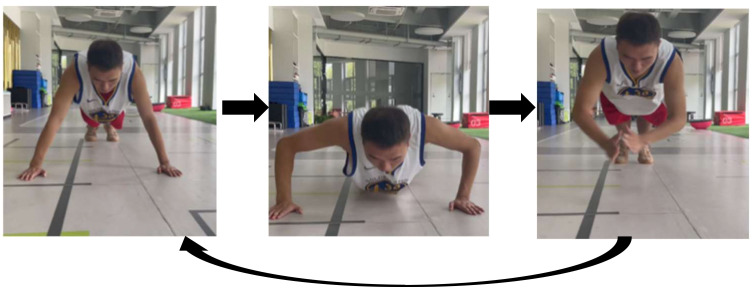
Schematic diagram of high-five push-ups.

**Figure 3 sensors-24-04439-f003:**
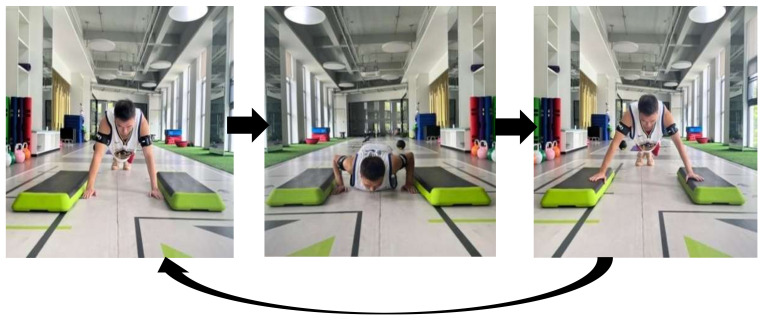
Schematic diagram of ballistic push-ups (when tied with a compression band).

**Figure 4 sensors-24-04439-f004:**
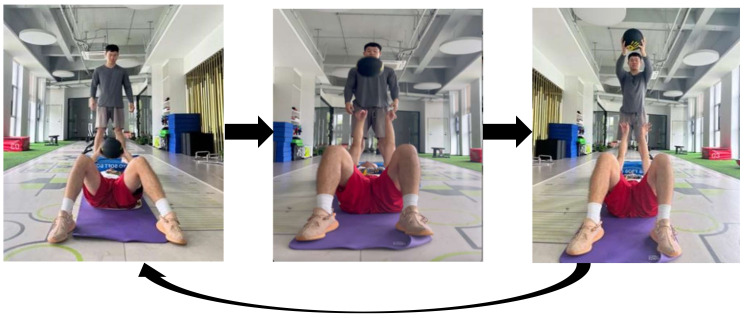
Schematic diagram of supine medicine ball forward push.

**Figure 5 sensors-24-04439-f005:**
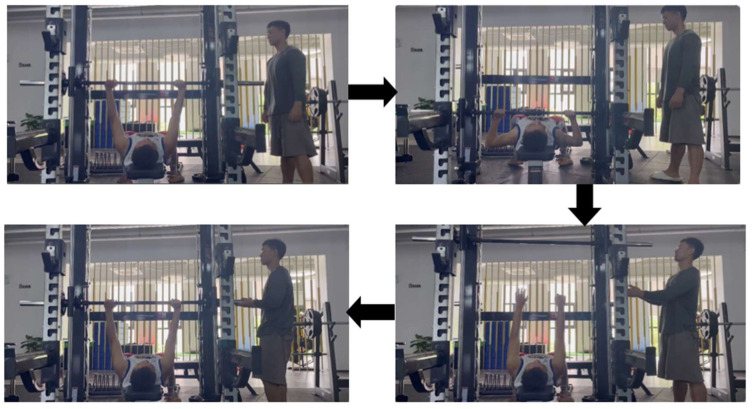
Schematic diagram of supine press and throw.

**Table 1 sensors-24-04439-t001:** List of subjects’ personal information.

Age (y)	Height (cm)	Body Mass (kg)	Arm’s Length	Training Period	Bench Press 1RM (kg)
23 ± 1.83	185.1 ± 5.49	81.7 ± 8.89	71.93 ± 2.57	3.27 ± 5.43	105.5 ± 11.65

**Table 2 sensors-24-04439-t002:** Results of two-factor repeated-measures ANOVA for changes in indicators in the BPT test.

	PV		MV		PP		MP	
	F	*p*	F	*p*	F	*p*	F	*p*
group	6.366	0.004 **	9.573	0.000 **	3.277	0.014 *	10.037	0.000 **
time	0.377	1.201	0.364	0.923	1.277	0.416	1.516	0.324
group × time	2.790	0.041 *	2.148	0.059	1.612	0.130	1.705	0.150

Note: * Indicates a significant difference, *p* < 0.05; ** indicates a highly significant difference, *p* < 0.01; group represents STR, BFR, COMB, and CON; peak power (PP), peak velocity (PV), mean power (MP), and mean velocity (MV); and H is the height of the push-up height during BPT, and Vmxpro can measure the height of the barbell pushed by the subject during the BPT test, which is also one of the indicators reflecting the explosive force.

## Data Availability

Published in the figshare database, DOI: https://doi.org/10.6084/m9.figshare.24471712, accessed on 1 June 2023.

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
