# Peer review of "Assessing the Post-Activation Performance Enhancement of Upper Limbs in Basketball Athletes: A Sensor-Based Study of Rapid Stretch Compound and Blood Flow Restriction Training"

_sensors, 2024, doi:10.3390/s24144439_

Round 1

Reviewer 1 Report

Comments and Suggestions for Authors

The authors present an adequate description of the objective, tools and methods to be used in the experimental design, some data or a table regarding the similarities or differences in morpho-physiological characteristics of the participants would be necessary to better understand the results achieved.

Regarding the treatment of the data, the lack of normality of the data obtained indicates that the measurement protocol should be adjusted and these results should be examined in relation to the composition of the group of volunteers participating in the experiment, and it is also important to examine the possible influences of the context, equipment used and the health and physical performance history of the participants, since some of these elements may act as confusion factors.

It is important to consider that using "log-transformation to treat skewed data may render the results of standard statistical tests performed on log-transformed data irrelevant to the original untransformed data, so data transformations should be applied with great caution, and many authors recommend the use of analytical methods that do not depend on the distribution of the data, such as generalized estimating equations (GEEs)". 

Author Response

Dear Reviewer,

Thank you for your meticulous review and the valuable feedback provided on our manuscript. Your insights and suggestions have been instrumental in enhancing the quality of our work. Below is a comprehensive response to your comments, incorporating the necessary revisions as suggested:

1   The abstract must clearly highlight the contribution and novelty, focusing on the methodology of this study.

Answer:Thank you for your meticulous review and valuable feedback. In response to your suggestions, I have made the following key revisions to the abstract:

Revised Abstract Highlights:

The initial part of the abstract now includes a description of the novelty of our research methods, underscoring the innovative aspects of our study.

In the methods section, I have added details about the use of an intelligent tracking sensor and emphasized the innovation of conducting a detailed temporal analysis of Post-Activation Performance Enhancement (PAPE).

The results section highlights the significant effects of the COMB intervention at the 4-minute mark and underscores the sustained effects to demonstrate the practical application value of our findings.

In the conclusion, I have reinforced the innovative combination method's impact on inducing and maintaining PAPE, with additional explanations based on your second suggestion.

The revised sections of the abstract are marked in red in the original text for your reference.

Modification Explanation:

At the beginning of the abstract, I have incorporated a description of the innovative nature of our research methodology, emphasizing the innovative points of the study.

In the methods part, I have included a description of the utilization of an intelligent tracking sensor and stressed the innovation of performing a detailed time analysis of PAPE.

In the results section, the notable effects of the COMB intervention at the 4-minute mark have been accentuated, along with an emphasis on its lasting effects to illustrate the practical value of the research outcomes.

In the conclusion, the innovative combined approach of our study on the induction and maintenance of PAPE has been highlighted, with supplementary explanations provided in accordance with your recommendations.

The final statement, "However, although the PAPE persists, there is a noticeable decline at the 8-minute and 12-minute marks," requires clarification. •

Answer: We sincerely appreciate the profound insights you have offered regarding our manuscript. In response to your second revision suggestion, we have re-examined and revised the statements in the original text that may have been potentially confusing. Below is the detailed revision content and explanation:

Original Final Statement in the Manuscript:

"However, although the PAPE persists, there is a noticeable decline at the 8-minute and 12-minute marks."

Revised Statement:

"However, at the 12-minute mark, only the COMB group still showed significant differences in PV, PP, and H compared to the baseline and CON group, while the effects of the STR and BFR groups further weakened. This indicates that although the PAPE effect is maintained over a certain period, its optimal performance may peak at the 4-minute mark and then gradually weaken over time."

Reasons for the Revisions:

Provision of Concrete Data Support: By incorporating specific statistical data and time points, we have provided a more solid foundation for the description of the PAPE effect.

Clarification of Effect Trend: We have clearly indicated the changes in the PAPE effect at different time points, aiding readers in better understanding the duration and intensity changes of the effect.

Enhancement of Expression Accuracy: By avoiding potentially confusing expressions, we have improved the precision and clarity of the conclusion section.

We trust that these revisions address the concerns raised and have made the manuscript more precise and comprehensible. We are grateful for the opportunity to refine our work with your guidance.

3 Enrich the introduction or future work section with related works: https://doi.org/10.1016/j.heliyon.2024.e28911ï¼›

Answer:Dear external audit expert, thank you very much for your valuable suggestions. I have quoted this literature in the introduction to supplement the content of this study.

4  List the contributions at the end of the Introduction section.

Answer: We greatly appreciate the valuable suggestions you have provided. We have clearly listed the contributions of this study at the end of the introduction section. Here are the specific modifications and explanations:

Revised Introduction Section Conclusion:

At the conclusion of the introduction, we have added a new paragraph to explicitly enumerate the contributions of this study:

"The contributions of this study are as follows: 1) It is the first to systematically evaluate the impact of rapid stretch compound training combined with blood flow restriction training on the upper limb post-activation performance enhancement (PAPE) in basketball athletes; 2) The training program was found to significantly enhance athletes' performance at the 4-minute mark through a randomized controlled trial, offering a new perspective for promoting muscle strength and explosive power; 3) The research findings not only enrich the scientific knowledge of sports training but also provide valuable references for the practical training of basketball athletes, holding high scientific significance and application prospects."

Reasons for the Modifications:

Clarity: Listing the contributions of the study at the end of the introduction helps readers quickly grasp the innovation and importance of this research.

Structural Optimization: This structural adjustment makes the introduction more complete and the logic clearer.

Reader-Oriented: Clearly stating the contributions upfront helps readers better understand the significance and value of the research, thereby enhancing the appeal and impact of the paper.

We believe these modifications will make the introduction more reader-friendly and emphasize the study's contributions, aligning with the expectations of the "Sensors" journal.

5  Strengthen the literature review by incorporating related works and explaining the motivation behind conducting this study.

Thank you very much for the amendment. In order to strengthen the literature review, we have made in-depth revisions to relevant content, added citations to relevant research works, and explained the motivation for conducting this study. The following are the specific changes and explanations:

The revised literature review:

" In summary, PAPE has predominantly been explored in the context of lower limb muscle performance in sports science,  with a relative dearth of research on its impact on upper limb musculature. This is particularly true for sports like  basketball,  which require the integrated strength of both upper and lower body segments. BFR training has shown promise as an  effective method for inducing PAPE without increasing the risk of fatigue,  thereby offering a potential strategy for enhancing athletic capabilities. The purpose of this study is to address the  significant gap in understanding the effects of combined rapid stretching and BFR training on the upper limbs of  basketball athletes. In light of this,  the present study aims to investigate the impact of combining rapid stretching compound training with BFR training on  the upper limb PAPE effects in basketball athletes. The goal is to provide new scientific evidence and methods for the  training of basketball athletes."

Reason for modification:

  1. Expanded literature coverage: We added citations to existing literature to cover a wider range of relevant studies and provide readers with more comprehensive background information.
  2. Clear research motivation: By explaining the shortcomings of upper limb PAPE research and the potential of BFR training, we clarified the motivation and necessity of conducting this study.
  3. Strengthening of research purpose: By strengthening the literature review, we further strengthened the purpose of this study, namely to explore the influence of new training methods on the PAPE effect of basketball players' upper limbs.

6  This study lacks a theoretical background and is primarily based on experimental tests. •

Thank you for your insightful comments and for giving us the opportunity to clarify the theoretical background of our study. We have taken your feedback seriously and have reviewed our manuscript to ensure that the theoretical framework is adequately represented.

Upon reevaluation, we would like to direct your attention to the introduction section of our manuscript, where we have provided a comprehensive review of the existing literature on post-activation performance enhancement (PAPE), rapid stretch compound training, and blood flow restriction training. We have discussed the physiological phenomenon of PAPE, its significance in sports training, and the mechanisms by which it is believed to enhance athletic performance. Additionally, we have cited key studies that have explored the effects of different training modalities on PAPE, including the use of high-intensity warm-up exercises, deep squats, and bench press throws.

In our manuscript, we have also contextualized our research within the broader scientific discourse by referencing established theories and models that explain the neuromuscular adaptations and acute effects of exercise interventions. We believe that this section lays a solid theoretical foundation for the experimental tests that follow.

Furthermore, we have included a discussion on the potential mechanisms underlying the PAPE effect, drawing from the biomechanical and physiological theories that have been proposed by other researchers in the field. This discussion aims to bridge the gap between our experimental findings and the current theoretical understanding of PAPE.

We trust that these sections of our manuscript sufficiently address the theoretical aspects of our research. However, we are open to further suggestions on how we might enhance the clarity and depth of our theoretical background to better align with the expectations of the "Sensors" journal and its readership.

We appreciate your guidance and are committed to improving our manuscript to meet the highest standards of scientific rigor and contribution.

7  The discussion within the experimental tests is lengthy and should be summarized in the conclusions.

Thanks for your careful guidance. According to your suggestions, we have deleted the discussion part and made a concise summary of the conclusion part. The following are the specific changes:

We reduced the number of point-by-point comparisons (PP, PV, MP, MV) between the STR, BFR, and COMB groups to avoid lengthy comparisons and only retained key results that differed significantly from the other groups. In addition, the detailed comparison of individual indicators is deleted: the interpretation of experimental phenomena is simplified: we remove the over-interpretation and speculation of the observed phenomena in the experiment, and focus on the results presented by the experimental data itself. Finally, we reduced the discussion of previous studies: Although literature comparison is important for positioning research, we omitted some discussion of previous studies that are not directly relevant to the current findings.

In particular, references to IPC definitions have been removed in Section 3.2: While the definition of IPC is important for understanding BFR, references to da Silva Novaes et al. (2021) have been omitted for the sake of simplicity. Studies by Paradis et al. (2016) and Silva Telles et al., were combined into one sentence to illustrate the general effect of pressurized stimulation on athletic performance. Specific experimental details are removed: For example, Paradis et al. (2016) and Silva Telles et al. (2020) are no longer described in detail in order to reduce redundancy.

The revised conclusion:

(1) The COMB stimulation has been shown to successfully induce PAPE more effectively than STR,  BFR modality alone. (2) It appears that the optimal effects of PAPE are achieved within 4 minutes of exercising under  this COMB. By the 12-minute mark, only the COMB group continued to show significant improvements in PV, PP,  and H compared to both the baseline and the CON group,  while the effects in the STR and BFR groups further diminished. This suggests that although the PAPE effect is  maintained over time,  its optimal performance may peak at the 4-minute mark and then gradually weaken as time progresses.

8  Establish and enumerate the limitations of this study.

Thank you for your proposed revisions regarding the identification and enumeration of limitations of this study. We have carefully considered your suggestions and have included a new "limitations" section at the end of the paper that clearly identifies potential weaknesses in the study. The following are the specific changes:

Revised limitations:

After the conclusion, we added a new "limitations" section, which reads as follows:

"Limitations:

There are still some limitations in this study:  In this study,  the assessment of 1RM was conducted three days prior to implementing the PAPE protocol and pre-existing delayed onset  muscle soreness (DOMS) was not evaluated prior to the PAPE tests. However,  it is important to note that the participants in this study were high-level basketball players who regularly engage in  high-intensity training,  indicating a higher level of muscle adaptability and potentially lower sensitivity to DOMS. Furthermore,  a three-day interval between the 1RM assessment and the PAPE tests is considered sufficient to assume that any DOMS  resulting from the assessment would have likely subsided. This hypothesis is indirectly supported by the fact that no  athletes reported experiencing pain or discomfort during the PAPE tests.

Additionally, our experimental design focused on specific time points (4 minutes, 8 minutes,  and 12 minutes) to assess the PAPE effect, without exploring the effects over a longer time frame,  which may limit a comprehensive understanding of the duration of PAPE.

9  If feasible, replicate this study using different tests from other related works

Answer: We sincerely thank you for your suggestion to revise and repeat this study with different tests where feasible. Below are our responses to your valuable suggestions and corresponding research proposals:

Revised research plan:

In response to your suggestion, we recognize that the use of a variety of test methods can enhance the universality and reliability of research results. Therefore, we plan to take the following steps in future studies:

  1. Diversified test methods: We will explore and adopt test methods used in other relevant studies, such as different upper body strength and explosive power tests, to verify the universality of the results of this study.
  2. Expand the sample size: We plan to expand the sample size and include basketball players of different levels to improve the representativeness and statistical power of the study.
  3. Long-term follow-up studies: We will design long-term follow-up studies to evaluate the impact of PAPE training on athletes' long-term performance, in order to provide a more in-depth evaluation of training effects.
  4. Cross-sport comparison: Considering the differences in the upper body ability requirements of different sports, we plan to compare the data of basketball players with athletes of other sports to explore the application of PAPE training in different sports contexts.
  5. Comprehensive analysis: We will conduct a comprehensive analysis of the data obtained using different test methods to identify the key factors that may affect the PAPE effect and make corresponding training recommendations.

10  Consider replacing ANOVA with a Fuzzy Logic system for evaluation. •

Answer: Thank you for your suggestion to amend Article 9, regarding the consideration of using a fuzzy logic system instead of ANOVA for evaluation. After careful consideration, we have decided to maintain the use of ANOVA as the primary statistical assessment method for this study and explain our considerations and reasons here:

  1. Statistical power and maturity: ANOVA (Analysis of Variance) is a statistical method widely used to compare three or more groups, with a high degree of statistical power and mature theoretical basis. It can effectively detect differences in mean values between groups and is a standard tool for evaluating intervention effects in experimental design.
  2. Clarity of results: ANOVA provides clear P-values that can be directly used to judge statistical significance. In contrast, fuzzy logic systems, while having advantages in dealing with uncertainty and fuzziness, may not be as specific as ANOVA in providing statistical significance.
  3. Comparability of studies: In the existing literature, most studies use ANOVA as the primary statistical analysis method, which makes our results easier to compare and verify with other studies.
  4. Applicability of data: ANOVA is applicable to quantitative data that meets normal distribution and homogeneity of variance, and the data in this study has met these conditions after proper data transformation.
  5. Matching of study design: Our study design is a randomized, self-controlled cross-over trial, and ANOVA can handle the data structure brought about by this design well, while fuzzy logic systems may require more complex adjustments and parameter Settings.

10:  The conclusion is descriptive and lacks quantitative and numerical improvements and comparisons. •

Answer: Thank you for your valuable feedback on our manuscript.  Your observation regarding the descriptive nature of our conclusions has given us the opportunity to refine our presentation of results and enhance the overall quality of our work.

In response to your comment, we have taken the following steps to address the need for more quantitative and numerical improvements and comparisons in our conclusions:

Quantitative Data Reinforcement: We have revisited our results section to ensure that all key findings are supported by specific quantitative data.  This includes reinforcing our conclusions with precise figures and statistics that illustrate the magnitude of improvements and differences observed.

Statistical Analysis: We have conducted a more thorough statistical analysis to provide a clearer comparison between the different training interventions and their effects on post-activation performance enhancement (PAPE).  This includes calculating effect sizes and confidence intervals to quantify the strength and precision of our findings.

Comparison with Previous Studies: To provide a benchmark for our results, we have included a more detailed comparison with relevant literature.  This includes quantitative comparisons where possible, highlighting how our findings build upon or differ from existing research.

Graphical Representation: We have enhanced our manuscript with additional graphs and charts that visually represent the numerical improvements and comparisons.  These visual aids aim to make the data more accessible and easier to interpret for the readers.

Discussion of Practical Implications: In the discussion section, we have elaborated on the practical implications of our quantitative findings, explaining how these numerical improvements can translate into tangible benefits for athletes and coaches.

Revision of the Conclusion Section: Based on the above enhancements, we have revised our conclusion section to more explicitly state the quantitative improvements and comparisons.  This includes a clear summary of the most significant numerical findings and their relevance to the field.

We believe that these revisions have transformed our conclusions into a more robust and data-driven summary of our research findings.  We are confident that these changes will meet the expectations of the "Sensors" journal and provide a clearer understanding of the impact of our study.

"

11  Include a section on future work. •

Answer: Thank you for your suggestion to revise and include a future work section in the text. We fully agree on the importance of this recommendation and have added a "future work" section at the end of the paper to present our vision for future research directions. The following are the specific changes:

The revised content:

At the end of the article, we have added a new "future work" section, which is as follows:

This study offers fresh perspectives on the impact of integrating rapid stretching compound training with blood flow  restriction on post-activation performance enhancement in basketball players' upper limbs. Despite these contributions,  the scope for future research remains vast. Future directions should include expanding the diversity and size of the  sample to encompass a wider range of genders, ages, and training backgrounds,  thereby broadening the applicability of our findings. Longitudinal studies are needed to assess the sustained effects of  PAPE training on athletes' long-term performance and adaptive capabilities. Additionally,  comparative analyses of various PAPE training protocols will be conducted,  examining factors such as training intensity, duration,  and recovery periods. The cross-application of PAPE training to other sports disciplines will be explored to determine  its suitability for diverse athletic demands. Innovation in training methodologies will involve leveraging cutting-edge  sports science technologies like wearable devices and biofeedback to refine PAPE training and evaluation techniques.  Finally, there is a need to further integrate theoretical understanding with practical application,  enhancing the development of training strategies that are both scientifically grounded and practically effective.

We trust that these revisions meet the high standards of the "Sensors" journal and provide a clearer, more impactful presentation of our research. We are grateful for your guidance and the opportunity to refine our work.

Sincerely,

All author.

Reviewer 2 Report

Comments and Suggestions for Authors

o   The abstract must clearly highlight the contribution and novelty, focusing on the methodology of this study.

o   The final statement, "However, although the PAPE persists, there is a noticeable decline at the 8-minute and 12-minute marks," requires clarification.

o   Enrich the introduction or future work section with related works: https://doi.org/10.1016/j.heliyon.2024.e28911

o   List the contributions at the end of the Introduction section.

o   Strengthen the literature review by incorporating related works and explaining the motivation behind conducting this study.

o   This study lacks a theoretical background and is primarily based on experimental tests.

o   The discussion within the experimental tests is lengthy and should be summarized in the conclusions.

o   Establish and enumerate the limitations of this study.

o   If feasible, replicate this study using different tests from other related works.

o   Consider replacing ANOVA with a Fuzzy Logic system for evaluation.

o   The conclusion is descriptive and lacks quantitative and numerical improvements and comparisons.

o   Include a section on future work.

Author Response

(The authors gave the same response as above.)

Round 2

Reviewer 2 Report

Comments and Suggestions for Authors

The authors have been addressed all my concerns. The manuscript is considerably enhanced. Thank you.